# AEQA-NAT: Adaptive End-to-end Quantization Alignment Training Framework for Non-autoregressive Machine Translation

Xiangyu Qu[1 2]   Guojing Liu[3]   Liang Li[1 2 4]

## Abstract

Non-autoregressive Transformers (NATs) have garnered significant attention due to their efficient decoding compared to autoregressive methods. However, existing conditional dependency modeling schemes based on masked language modeling introduce a *training-inference gap* in NATs. For instance, while NATs sample target words during training to enhance input, this condition cannot be met during inference, and simply annealing the sampling rate to zero during training leads to model performance degradation. We demonstrate that this *training-inference gap* prevents NATs from fully realizing their potential. To address this, we propose an adaptive end-to-end quantization alignment training framework, which introduces a semantic consistency space to adaptively align NAT training, eliminating the need for target information and thereby bridging the *training-inference gap*. Experimental results demonstrate that our method outperforms most existing fully NAT models, delivering performance on par with Autoregressive Transformer (AT) while being 17.0 times more efficient in inference.

## 1. Introduction

Non-autoregressive Transformer (NAT, Gu et al. 2018) has emerged as a promising approach to mitigate the high latency inherent in Autoregressive Transformer (AT, Vaswani et al. 2017), which stems from their sequential token-by-token decoding mechanism. While NAT achieves significant speedup by parallel decoding, it often suffers from translation quality degradation due to insufficient modeling of

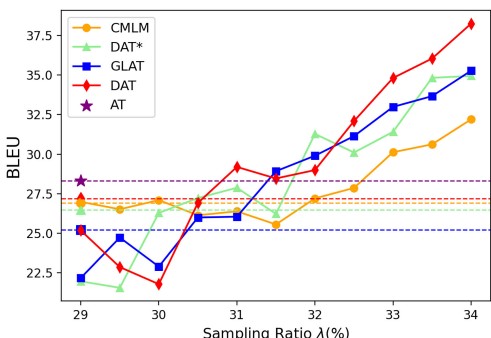

*Figure 1.* An example illustrating the challenge of the training-inference gap faced by MLM-based Non-autoregressive Transformer (NAT) models. In this context, the x-axis represents the sampling rate of unmasked target tokens during the inference phase of the NAT model, which is consistent with the training phase. The y-axis denotes the translation quality. The dashed line indicates the baseline performance of the NAT model under its original inference conditions.

interdependicies among target tokens (Guo et al., 2019; Xiao et al., 2023). Current research in NAT primarily explores two paradigms: iterative decoding, which refines translations through multiple iterations to balance quality and efficiency (Ghazvininejad et al., 2020b; Saharia et al., 2020), and non-iterative decoding, which generates translations in a single forward pass to maximize speed advantages (Gu & Kong, 2021; An et al., 2023).

Conditional masked language modeling (CMLM) stands as one of the most effective approaches for enhancing target token dependencies in NAT (Ghazvininejad et al., 2019; Shao & Feng, 2022; Huang et al., 2022d). This method explicitly guides the model to learn the mapping from the source sentence $X$ and observable target tokens $Y_{obs}$ to the masked target tokens $Y_{mask}$ by incorporating additional target words as part of the input. Qian et al. (2021) further advanced this explicit dependency modeling capability by introducing the Glancing Transformer (GLAT), which adaptively adjusts the number of sampled target tokens, a technique that has been widely adopted (Bao et al., 2022; Huang et al., 2022c; Guo et al., 2023). However, this MLM-based training paradigm $(X + Y_{obs} \rightarrow Y_{mask})$ creates a training-inference gap, as

---

[1]School of Cyber Science and Technology, Shandong University [2]State Key Laboratory of Cryptography and Digital Economy Security, Shandong University [3]Ocean University of China [4]QCL. Correspondence to: Liang Li <li.liang@sdu.edu.cn>.

*Proceedings of the 42nd International Conference on Machine Learning*, Vancouver, Canada. PMLR 267, 2025. Copyright 2025 by the author(s).

NAT inference requires predicting the complete translation $(X \rightarrow Y)$ rather than reconstructing partial tokens. While annealing the target token sampling rate to zero during training might theoretically address this gap, GLAT demonstrates that doing so leads to significant performance degradation.

The state-of-the-art NAT model, Directed Acyclic Transformer (DAT, Huang et al. 2022c), optimizes translation paths by constructing a directed acyclic graph, achieving performance comparable to AT models without knowledge distillation. Nevertheless, DAT is not without its limitations: 1) its performance gains come at the expense of decoding speed, with speedups ranging from 14.0x to 7.0x (Huang et al., 2022c); and 2) despite employing GLAT training methods, DAT fails to fully completely bridge the gap between training and inference. Furthermore, empirical investigations into the unified training and inference of NAT—where target information is input during inference to prompt translation—reveal significant improvements in translation quality for MLM-based NAT models, as shown in Fig 1. This indicates that eliminating the training-inference gap can significantly enhance model performance.

Our main contributions are as follows:

- Our research has revealed a significant discrepancy between the training and inference of contemporary advanced NAT networks. Specifically, we've identified that NAT fails to reach its full potential during inference.

- We propose an Adaptive End-to-End Quantization Alignment (AEQA) framework for NAT, which introduces a semantic consistency space to jointly optimize all model components without requiring additional target information. This approach effectively eliminates the training-inference gap inherent in NAT systems.

- Experimental results demonstrate that our method achieves the fastest decoding speed among all NAT networks while attaining state-of-the-art performance across multiple translation directions. Additionally, our method reduces the performance gap between training on raw data and distilled data to 0.29 BLEU points, demonstrating its superiority in handling multimodal distributions.

## 2. Methodology

In this section, we first systematically introduce the widely used techniques of existing advanced NAT models and reveal their limitations. We then present our AEQA-NAT framework in detail, including its training and inference process with some effective schemes.

### 2.1. Preliminary

**Non-autoregressive Machine Translation** The machine translation task can be formally defined as a sequence-to-sequence generation problem, where the neural network generates a target sentence $Y = \{y_1, y_2, \cdots, y_N\}$ under the condition of a given source sentence $X = \{x_1, x_2, \cdots, x_M\}$. Autoregressive Transformer (AT, Vaswani et al. 2017) factorize the translation probability as follows and maximize it with the cross-entropy loss

$$\mathcal{L}_{\text{AT}} = -\sum_{i=1}^{N} \log p_\theta(y_i|y_{<i}, X) \qquad (1)$$

where $y_i$ is predicted based on the prefix $y_{<i}$.

The Vanilla NAT (Gu et al., 2018) makes a conditional independent assumption where each token is independent of each other when $X$ is given. Formally, we have

$$\mathcal{L}_{\text{NAT}} = -\sum_{i=1}^{N} \log p_\theta(y_i|X) \qquad (2)$$

All target tokens are generated in parallel through the conditional probability of $X$. During decoding, NAT lacks appropriate methods to restore correct inter-word dependencies.

**The Conditional Masked Language Model** Enhancing dependency modeling is based on the idea of masked language model (MLM), e.g., CMLM (Ghazvininejad et al., 2019), GLAT (Qian et al., 2021) and DAT (Huang et al., 2022c), which allows the model to explicitly learn a mapping from the source sequence $X$ and the observable variables [1] $Y_{obs}$ (unmasked ground truth tokens) to the target sequence $y$

$$\mathcal{L}_{mask} = -\sum_{y \in Y_{mask}} \log p_\theta(y|Y_{obs}, X) \qquad (3)$$

$\mathcal{L}_{mask}$ denotes the reconstruction loss. MLM-based construction loss plays a critical role in enhancing dependency modeling of target tokens, which significantly improves the translation quality of NAT. However, the training paradigm $\langle X + Y_{obs} \rightarrow Y_{mask} \rangle$ makes NAT learn a revised distribution (Huang et al., 2022b), resulting in a gap between NAT training and inference (i.e., NAT inputs $X$ and predicts the full $Y$ in inference).

---

[1] In CMLM, the observable variables are derived from a random sampling $\mathbb{RM}(Y)$ of the ground truth $Y$, which is formulated as $\mathcal{L}_{\text{MLM}} = \sum_{y \in \mathbb{RM}(\mathbb{Y})} \log p_\theta(y|\mathbb{RM}(Y), X)$. In GLAT, the observable variables are obtained through the glancing sampling strategy $\mathbb{GS}(Y, \hat{Y}) = \text{Random}(Y, \mathcal{D}(\hat{Y}, Y))$, expressed as $\mathcal{L}_{\text{GLM}} = \sum_{y \in \mathbb{GS}(Y, \hat{Y})} \log p_\theta(y|\mathbb{GS}(Y, \hat{Y}), X)$.

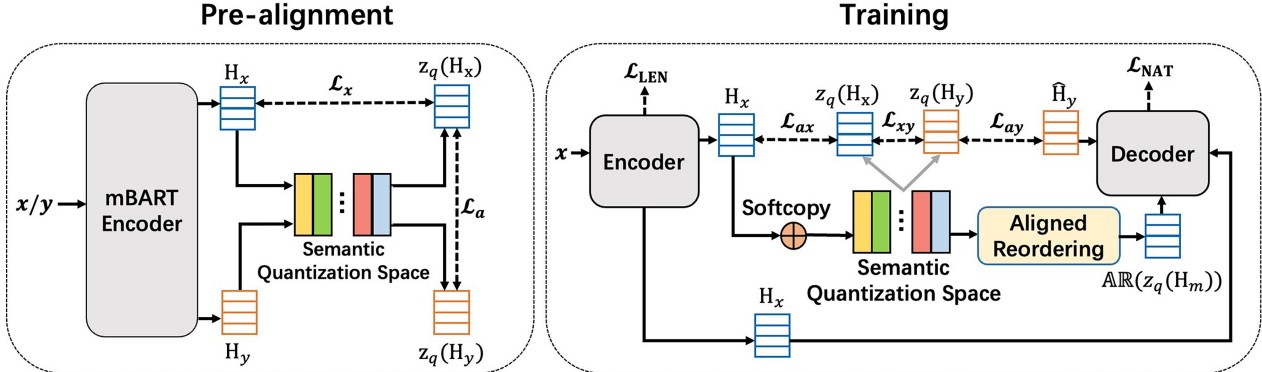

*Figure 2.* Overview of the AEQA training framework. In the pre-alignment phase, we utilize exponential moving average updates on the mBART model to refine the Semantic Quantization Space. In the training stage, we optimize the whole model through $\mathcal{L}_{\text{NAT}}$, $\mathcal{L}_{\text{SQA}}$, and $\mathcal{L}_{\text{LEN}}$.

## 2.2. Adaptive End-to-End Quantization Alignment

In contrast to Autoregressive Transformers, which employ a consistent left-to-right decoding strategy during both training and inference, NATs face incompatibility issues due to the additional information required during training that is unavailable at inference. Inspired by Vector Quantized Variational Autoencoder (VQ-VAE, Van Den Oord et al. 2017), we construct an external semantic quantization space (SQS) to serve as a bridge and constraint for the NAT system during training. This SQS is co-optimized with the model to ensure that the NAT system captures the semantic consistency between the source and target languages. By doing so, we achieve a unified framework for NAT systems across both training and inference phases. Existing approaches such as latent-GLAT (Bao et al., 2022) enhance NAT through latent variable modeling. Our method fundamentally diverges from such VAE-based NAT frameworks, as elaborated in Appendix A.1.

## 2.3. Architecture of AEQA Training Framework

As illustrated in Fig 2, the adaptive end-to-end quantization alignment training framework mainly consists of three modules: a *NAT encoder* $f_{\text{enc}}$, a shared *semantic quantization space* SQS and a *NAT decoder* $f_{\text{dec}}$. Their functions can be formalized as

$$(h_{x,1}, h_{x,2}, \cdots, h_{x,n}) \leftarrow f_{\text{enc}}(x_1, x_2, \cdots, x_n),$$
$$(h_1, h_2, \cdots, h_m) \leftarrow \text{softcopy}(f_{\text{len}}(h_{x,1}, h_{x,2}, \cdots, h_{x,n})),$$
$$(z_q(h_1), z_q(h_2), \cdots, z_q(h_m)) \leftarrow \text{SQS}(h_1, h_2, \cdots, h_m),$$
$$p_\theta(Y|\mathcal{SQ}(X), X) \leftarrow f_{\text{dec}}(z_q(h_{1:m}), h_{x,1:x,n}),$$

We use an extra module $f_{\text{len}}$ to predict the target length $m$ and initialize the decoder inputs $\mathbf{H}_m = \{h_1, h_2, \cdots, h_m\}$ with the *Softcopy* (Li et al., 2018; Wei et al., 2019) mechanism, see Appendix C.2 for more details.

**Pre-aligned Semantic Quantization Space** We leverage the pre-trained multilingual model mBART (Liu et al., 2020) to achieve the alignment between the source language and the target language within the SQS, as depicted in the Fig 2, mBART[2] is a pre-trained cross-lingual model based on 25 languages, and by relying on its shared encoder, it can effectively embed the source sequence and the target sequence into the SQS. Specifically, the SQS can be viewed as a $K \times$ D-dimensional vocabulary $S = \{e_1, e_2, ..., e_k\}$, where $k$ is the number of word embeddings and D is the dimension of the word embedding $e_k$. Intuitively, given the hidden states $\mathbf{H}_x = \{h_{x,1}, h_{x,2}, ..., h_{x,N_x}\}$, mapping each dimension of $\mathbf{H}_x$ to one of the $K$ embeddings contained in the SQS. Formally, we have

$$z_q(h_{x,i}) = \arg\min_{e_{k'}} \|h_{x,i} - e_{k'}\|_2 \quad (4)$$

$$z_q(h_{y,j}) = \arg\min_{e_{k''}} \|h_{y,j} - e_{k''}\|_2 \quad (5)$$

where $k'$ denotes the index of SQS vector $e'_k$ that minimizes the distance $\|h_{x,i} - e_{k'}\|_2$, effectively quantizing $h_{x,i}$ into the category represented by $e'_k$.

Following (Van Den Oord et al., 2017), we constrain the outputs of the encoder and decoder to the vectors in SQS with commitment loss (i.e., $\mathcal{L}_x$). Concretely, we use exponential moving average (EMA) to update SQS

$$w_k = \sum_{i}^{N_x} \mathbb{I}((h_{x,i}) = k)h_{x,i}$$

$$n_k \leftarrow \gamma n_k + (1 - \gamma) \sum_{i}^{N_x} \mathbb{I}((h_{x,i}) = k) \quad (6)$$

$$e_k \leftarrow \frac{1}{n_k}(\gamma e_k + (1 - \gamma)w_k)$$

---

[2]https://github.com/facebookresearch/fairseq/tree/main/examples/mbart

where $\mathbb{I}(\cdot)$ is the indicator function, $w_k$ is computed as the sum of the encoder's hidden states $h_{x,i}$, and $\gamma$ is a decay factor. $e_k$ is updated by averaging the previous embedding and the newly computed $w_k$, normalized by $n_k$. For more details on constructing the SQS, refer to Appendix A.2.

## 2.4. Training

AEQA-NAT optimizes two training objectives: semantic quantization alignment loss $\mathcal{L}_{\text{SQA}}$ and translation maximum likelihood loss $\mathcal{L}_{\text{NAT}}$.

**Semantic Quantization Alignment Loss** $\mathcal{L}_{\text{SQA}}$ We leverage the pre-aligned SQS to facilitate the model's semantic mapping from the source sequence to the target sequence, thereby ensuring the consistency of the encoder and decoder outputs within the SQS during the training process. Specifically, we have

$$\mathcal{L}_{\text{SQA}} = \mathcal{L}_{\text{ax}} + \mathcal{L}_{\text{xy}} + \mathcal{L}_{\text{ay}} \tag{7}$$

$$\mathcal{L}_{\text{ax}} = \|\mathbf{H}_x - \text{sg}(z_q(\mathbf{H}_x))\|_2^2 \tag{8}$$

$$\mathcal{L}_{\text{xy}} = \|z_{\bar{q}}(\mathbf{H}_x) - \text{sg}(z_{\bar{q}}(\mathbf{H}_y))\|_2^2 \tag{9}$$

$$\mathcal{L}_{\text{ay}} = \|\hat{\mathbf{H}}_y - \text{sg}(z_q(\mathbf{H}_y))\|_2^2 \tag{10}$$

where $\hat{\mathbf{H}}_y$ represents the hidden representation output by the last layer of the decoder, which is consistent with the dimension of SQS. Note that $z_{\bar{q}}(\mathbf{H}_x)$ is calculated as $\frac{1}{N_x} \sum_{i=1}^{N_x} z_q(h_{x,i})$ and $z_{\bar{q}}(\mathbf{H}_y)$ is calculated as $\frac{1}{N_y} \sum_{j=1}^{N_y} z_q(h_{y,j})$.

**Translation Maximum Likelihood Loss** $\mathcal{L}_{\text{NAT}}$ To ensure the model generates complete translations while preserving the benefits of Glancing training, we design a learning strategy compatible with Glancing Targets (Qian et al., 2021). Specifically, $\mathcal{L}_{\text{NAT}}$ maximizes the conditional probability of the target sequence $Y$ given the input sequence $X$:

$$\mathcal{SQ}(X) = \mathbb{GS}(\text{SQS}(f_{enc}(X)), \mathcal{D}(Y, \hat{Y})) \tag{11}$$

$$\mathcal{L}_{\text{NAT}} = -\sum_{i=1}^{T} \log p_\theta(y_i | \mathcal{SQ}(X), X) \tag{12}$$

where $\mathbb{GS}(\cdot)$ denotes Glancing Sampling Strategy (Qian et al., 2021), $\mathcal{D}(Y, \hat{Y})$ is used to determine the number of samples[3]. Note that what is replaced here is the discrete vector, rather than the target tokens. $\mathcal{SQ}(\cdot)$ is the quantized input after masking. Finally, the overall training loss $\mathcal{L}$ is obtained by

$$\mathcal{L} = \mathcal{L}_{\text{NAT}} + \epsilon \mathcal{L}_{\text{SQA}} + \delta \mathcal{L}_{\text{LEN}} \tag{13}$$

where $\delta$ and $\epsilon$ are the hyperparameters to control the impact of Semantic Quantization Alignment loss $\mathcal{L}_{\text{SQA}}$ and length

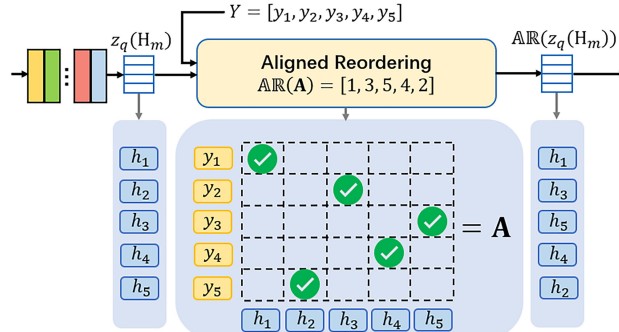

*Figure 3.* Aligned Reordering (AR) process applied to a sequence of semantic representations. The input sequence $z_q(\mathbf{H}_m) = \{h_1, h_2, ..., h_m\}$ is aligned with the ground truth sequence $Y = \{y_1, y_2, ..., y_n\}$. The alignment probability distribution matrix $\mathbf{A}$ is computed to determine the most probable order the input words. The AR process outputs the reordered sequence $\mathbb{AR}(z_q(\mathbf{H}_m)) = \{h_1, h_3, h_5, h_4, h_2\}$, aligning the input sequence with the correct syntactic structure as indicated by the matrix $\mathbf{A}$. The checked positions in the matrix represent the optimal alignment of each word in the sequence.

prediction loss $\mathcal{L}_{\text{LEN}}$, respectively. Based on this, gradients are computed to update various parameters of the model, including the encoder parameters, decoder parameters, and the embedding vectors $e_k$ of the SQS.

**Aligned Reordering** Intuitively, the input to the decoder, $\mathcal{SQ}(X)$, is strongly correlated with the word order of the source sequence rather than the target sequence. These representations may not initially adhere to the correct syntactic order with respect to the target language. The Aligned Reordering (AR) mechanism[4] addresses this by adjusting the syntactic discrete representations $z_q(\mathbf{H}_m)$ from the SQS, corresponding to words in the source sequence, as illustrated in Fig 3. Specifically, we define an alignment probability distribution matrix A. Given the input $z_q(\mathbf{H}_m) = \{h_1, h_2, ..., h_m\}$ from the SQS and the ground truth $Y = \{y_1, y_2, ..., y_n\}$, the alignment probability distribution matrix A is computed as follows

$$\mathbf{A} = \text{softmax}(Y z_q(\mathbf{H}_m)^T) \tag{14}$$

where $\mathbf{A} \in \mathbb{R}^{n \times m}$ is the alignment probability distribution matrix normalized by rows, which captures the alignment probabilities between the input representation $z_q(\mathbf{H}_m)$ and the ground truth $Y$. The AR process uses these probabilities to reorder the input sequence into the correct syntactic order, as indicated by the reordered sequence $\mathbb{AR}(z_q(\mathbf{H}_m)) = \{h_1, h_3, h_5, h_4, h_2\}$. The detailed formulation is presented in Appendix C.1.

---

[3]The sampling number is computed as $t = \lambda \sum_{i=1}^{T} (y_i \neq \hat{y}_i)$, where the sampling ratio $\lambda \in [0, 1]$ is a hypermeter.

[4]Note that Aligned Reordering adjusts the word order without altering the word vectors themselves.

*Table 1.* Performance comparison between our models and existing methods. The speedup is measured on WMT 14 EN↔DE test set with batch size 1. $I_{dec}$ denotes the number of iterations at inference time, Adv means adaptive and $m$ is the number of length reranking candidates. Results of prior work are quoted from respective papers. NPD represents noisy parallel decoding. Reordering denotes the aligned reordering mechanism. DSLP (Huang et al., 2022a) denotes deep supervision and feed additional layer-wise predictions. Best performance of non-iterative NATs ($I_{dec}$=1) are **bolded**. * indicates results of our re-implementation. † denotes the results of out implementations.

| Model | $I_{dec}$ | WMT14 EN-DE Raw | KD | WMT14 DE-EN Raw | KD | WMT16 EN-RO Raw | KD | WMT16 RO-EN Raw | KD | Speedup |
|---|---|---|---|---|---|---|---|---|---|---|
| Transformer (Vaswani et al., 2017) | M | 27.37 | 27.48 | 31.33 | 31.34 | 33.89 | 33.65 | 34.12 | 34.00 | 1.0x |
| Transformer (Ours) | M | 28.11 | 28.37 | 31.90 | 31.88 | 33.70 | 33.68 | 34.39 | 34.05 | 1.0x |
| CMLM (Ghazvininejad et al., 2019) | 10 | 24.61 | 27.03 | 29.40 | 30.53 | 32.86 | 33.08 | - | 33.31 | 1.7x |
| JM-NAT (Guo et al., 2020) | 10 | - | 27.69 | - | 32.24 | - | 33.52 | - | 33.72 | 5.7x |
| SMART (Ghazvininejad et al., 2020b) | 10 | 25.10 | 27.65 | 29.58 | 31.27 | - | - | - | - | 2.2x |
| DisCo (Kasai et al., 2020) | Adv | - | 27.34 | - | 31.31 | - | 33.22 | - | 33.25 | 2.6x |
| Multi-Task NAT (Hao et al., 2021) | 10 | 25.79 | 27.98 | 30.32 | 31.27 | - | 33.80 | - | 33.60 | 2.6x |
| RewriteNAT (Geng et al., 2021) | Adv | - | 27.83 | - | 31.52 | - | 33.63 | - | 34.09 | - |
| CMLMC (Huang et al., 2022d) | 10 | 26.40 | 28.37 | 30.92 | 31.41 | 34.14 | 34.57 | 34.13 | 34.14 | - |
| Con-NAT (Cheng & Zhang, 2022) | 10 | 25.60 | 27.93 | 30.05 | 31.57 | - | 33.88 | - | 34.18 | - |
| Vanilla NAT (Gu et al., 2018) | 1 | 10.84* | 18.21* | 15.85* | 24.33* | 19.82* | 27.29 | 21.93* | 29.06 | 15.6x |
| CTC (Libovický & Helcl, 2018) | 1 | 17.73* | 25.52 | 21.44* | 28.73 | 23.12* | 32.60 | 25.05* | 33.46 | 14.6x |
| AXE (Ghazvininejad et al., 2020a) | 1 | 20.40 | 23.53 | 24.90 | 27.90 | 30.47 | 30.75 | 31.42 | 31.54 | 14.5x |
| GLAT (Qian et al., 2021) | 1 | 18.94* | 25.21 | 25.71* | 29.84 | 26.38* | 31.19 | 27.99* | 32.04 | 14.2x |
| OaXE (Du et al., 2021) | 1 | 22.40 | 26.10 | 26.80 | 30.20 | 25.43* | 32.40 | 28.17* | 33.30 | 12.4x |
| DAT (Huang et al., 2022c) | 1 | 26.57 | 27.49 | 30.68 | 31.37 | 32.71* | 32.79* | 33.25* | 33.85* | 13.9x |
| MgMO (Li et al., 2022) | 1 | - | 26.40 | - | 30.30 | - | 32.90 | - | 33.60 | - |
| DePA (Zhan et al., 2023) | 1 | - | 26.43 | - | 30.42 | - | 33.07 | - | 33.82 | 15.1x |
| Renew NAT (Guo et al., 2023) | 1 | - | 26.65 | - | 30.65 | - | 33.02 | - | 33.74 | 11.2x |
| FA-DAT (Ma et al., 2023) | 1 | **27.47** | 27.17 | **31.44** | - | - | - | - | - | 13.2x |
| CMLM-rephraser (Shao et al., 2023) | 1 | 23.12 | 26.65 | 27.44 | 30.70 | 32.30 | 32.72 | 32.07 | 33.03 | 15.0x |
| DAT* (Li et al., 2024) | 1 | 26.48* | 27.01* | 30.62* | 31.15* | 33.18 | 33.25 | 33.02* | 33.14* | 12.0x |
| †AEQA-NAT | 1 | 25.96 | 26.24 | 28.78 | 29.04 | 30.83 | 31.11 | 31.29 | 31.67 | 17.0x |
| †AEQA-NAT w/ Reordering | 1 | 26.82 | 27.04 | 29.61 | 29.85 | 32.75 | 32.80 | 32.07 | 32.11 | 17.0x |
| †AEQA-NAT w/ NPD (m=5) | 1 | 26.87 | 27.20 | 30.14 | 30.53 | 32.78 | 32.79 | 32.25 | 32.59 | 13.5x |
| †+DSLP | 1 | 26.22 | 26.29 | 28.95 | 29.14 | 28.97 | 29.16 | 32.01 | 32.34 | 15.3x |
| †AEQA-NAT w/ Reordering | 1 | 26.95 | 27.21 | 30.25 | 30.34 | 32.80 | 32.83 | 32.96 | 33.27 | 15.3x |
| †AEQA-NAT w/ NPD (m=5) | 1 | 27.10 | 27.30 | 30.71 | 31.42 | **33.26** | 33.31 | 33.42 | 33.76 | 12.2x |
| †AEQA-DAT | 1 | 27.03 | **27.62** | 30.94 | **31.70** | 33.09 | **33.37** | **33.56** | **33.89** | 9.1x |

## 2.5. Inference

AEQA-NAT utilizes the source text and the quantization alignment representations during inference, which is consistent with the training process. Specifically, we obtain

$$\hat{Y} = \arg\max_{Y} \log p_\theta(Y|\mathcal{SQ}(X), X) \qquad (15)$$

benefiting from the fact that no additional target information is introduced during training, AEQA-NAT seamlessly generalizes the knowledge from the training data to the inference stage[5].

---
[5]During inference, $\mathcal{SQ}(\cdot)$ utilizes a fixed sampling rate.

## 3. Experiments

**Dataset** We validate our proposed models on four widely used translation benchmarks, i.e., WMT14 EN↔DE (4.0M), WMT16 EN↔RO (610K), WMT17 ZH↔EN (20M) and IWSLT16 DE→EN (153K), where we follow (Zhou et al. 2020, Lee et al. 2018a, Kasai et al. 2020) for pre-processing. Consistent with previous work (Gu et al. 2018, Qian et al. 2021), we employ sequence-level knowledge distillation for all datasets, refer to Appendix B.

**Evaluation** Following prior works, we compute tokenized BLEU (Papineni et al., 2002) for WMT14 EN↔DE and WMT16 EN↔RO, while using SacreBLEU (Post, 2018) for WMT17 ZH↔EN.To comprehensively assess translation quality, we utilize multiple additional metrics, including the rule-based metric chrF (Popović, 2015) and two model-

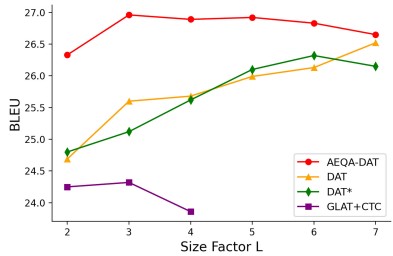

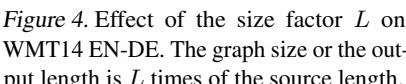

Figure 4. Effect of the size factor $L$ on WMT14 EN-DE. The graph size or the output length is $L$ times of the source length.

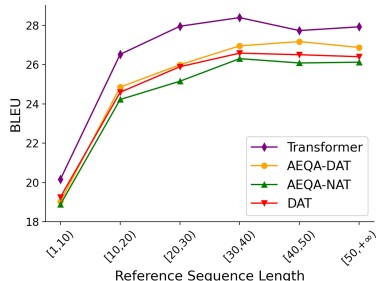

Figure 5. The BLEU score on WMT14 EN-DE bucketed by the reference length.

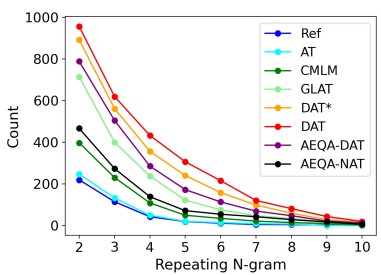

Figure 6. N-gram Repetition of different models.

based metrics: COMET (Rei et al., 2020) and BLEURT (Sellam et al., 2020). Specifically, for COMET, we use the wmt22-comet-da model (Rei et al., 2022), and for BLEURT, we adopt the BLEURT-20 model (Pu et al., 2021).

**Implementations** Our models generally use the hyperparameters of transformer-*base* (Vaswani et al., 2017). We set the size of SQS to 2048. During pre-aligning, we set $\alpha$ in Eq (16) to 0.5. We set the dropout rate to 0.1 and use Adam optimizer (Kingma & Ba, 2014) with $\beta = (0.9, 0.999)$. During training, we set $\epsilon$ and $\delta$ in Eq (13) to 0.25 and 0.1, respectively. We apply weight decay 0.01 and label smoothing $l = 0.1$. We train the model with the batches of 64K/8K tokens for WMT/IWSLT datasets, respectively. The learning rate warms up to $5e - 4$ in 4K steps and gradually decays according to inverse square root schedule. For hyparameter $\lambda$, we adopt linear annealing from 0.6 to 0.4 for training and a fixed value of 0.4 in inference. We apply noisy parallel decoding denoted as NPD (Gu et al., 2018) and set the length beam as 5. We extend our method to DAT (Huang et al., 2022c) by setting the graph size factor $L$ and removing the length prediction module, refer to Appendix D. All models are implemented on `fairseq` (Ott et al., 2019).

## 4. Main Results

The main results on the benchmarks are presented in Table 1, AEQA-NAT demonstrates significant advantages in translation performance. Our method enhances conditional dependency modeling at the decoder through the Semantic Quantization Space (SQS), effectively eliminating the training-inference gap and enabling the model to achieve its full potential. The SQS ensures that the model can directly learn complex raw data distributions, thereby providing a promising solution to eliminate the reliance on knowledge distillation.

1) In terms of translation quality, AEQA-NAT achieves state-of-the-art results across multiple translation directions, demonstrating significant performance advantages compared to fully NAT models. Compared to AT, AEQA-NAT exhibits a performance gap of less than 1 BLEU across all

Table 2. Results of AT and NAT models trained with (or without) knowledge distillation on WMT14 EN↔DE and IWSLT16 DE→EN.

| Methods | WMT14 | | IWSLT16 | Avg Gap $\Delta \downarrow$ |
|---|---|---|---|---|
| | EN→DE | DE→EN | DE→EN | |
| Vanilla NAT | 10.84 | 15.85 | 17.93 | +7.24 |
| w/ KD | 18.21 | 24.33 | 23.81 | |
| GLAT | 18.94 | 25.71 | 29.20 | +4.21 |
| w/ KD | 25.21 | 29.84 | 31.43 | |
| DAT | 26.57 | 30.68 | 31.57 | +0.81 |
| w/ KD | 27.49 | 31.37 | 32.40 | |
| AEQA-NAT | 26.87 | 30.14 | 32.34 | +0.29 |
| w/ KD | 27.20 | 30.53 | 32.50 | |
| Transformer | 28.11 | 31.90 | 32.92 | **+0.03** |
| w/ KD | **28.37** | **31.88** | **32.78** | |

benchmarks, surpassing other NAT models and indicating its ability to generate translations that are semantically and syntactically more aligned with the target language.

2) In terms of decoding speed, AEQA-NAT shows a clear advantage over AT and iterative NAT models, achieving a maximum speedup of 17.0x, which is also superior to other fully NAT models. Even when integrated with NPD techniques, AEQA-NAT maintains a high decoding speed with a speedup ratio of 12.2x, retaining its edge over other NAT models. This demonstrates that AEQA-NAT achieves an effective balance between translation quality and decoding efficiency.

3) In terms of portability, AEQA-NAT exhibits exceptional performance. It can seamlessly integrate advanced NAT techniques, such as NPD and DSLP, without significantly increasing computational overhead. This highlights the efficiency and flexibility of the AEQA-NAT framework.

### 4.1. Analysis

**AEQA Enhances Dependency Modeling on Raw Data**
Knowledge distillation (KD) is widely employed to enhance NAT learning. Distilled data constitutes a mixed marginal

*Table 3.* Performances on WMT14 EN→DE and WMT17 ZH→EN with the fixed sampling ratio in inference.

| Sampling ratio $\lambda$ | 0.1 | 0.2 | 0.3 | 0.4 | 0.5 | 0.6 |
|---|---|---|---|---|---|---|
| WMT14 EN-DE | 18.34 | 20.71 | 24.53 | **26.82** | 25.64 | 23.87 |
| WMT17 ZH-EN | 12.96 | 15.37 | 19.11 | 23.18 | **23.26** | 20.95 |

*Table 4.* Results on WMT14 EN→DE test sets with different number of references (e.g., "Single" and "Multiple"). "Δ" indicates the performance gap over the Vanilla NAT.

| Methods | Single | | Multiple | |
|---|---|---|---|---|
| | BLEU | Δ | BLEU | Δ |
| **Raw Data** | | | | |
| Vanilla NAT | 10.8 | - | 25.0 | - |
| CMLM | 11.0 | +0.2 | 28.1 | +3.1 |
| GLAT | 18.9 | +8.1 | 51.5 | +26.5 |
| AEQA-NAT | 26.8 | +16.0 | 71.3 | **+46.3** |
| **Distillation** | | | | |
| Vanilla NAT | 15.9 | - | 41.9 | - |
| CMLM | 18.6 | +2.7 | 50.7 | +8.8 |
| GLAT | 25.2 | +9.3 | 65.3 | +23.4 |
| AEQA-NAT | 27.0 | +11.1 | 73.5 | **+31.6** |

distribution derived from raw data and the teacher model's distribution. As illustrated in Table 2, the Transformer's translation quality improves by merely 0.03 BLEU after applying KD, underscoring its robust ability to model raw data and minimal dependence on distilled data. Intuitively, a model's capacity to capture data distribution characteristics inversely correlates with its reliance on distilled data. Vanilla NAT shows significant dependence on distilled data, achieving a BLEU score improvement of 7.24 post-KD. DAT markedly reduces this gap to 0.81, while AEQA-NAT achieves a further reduction to 0.29 for the first time. These results demonstrate that AEQA effectively captures raw data distribution features and exhibits advanced dependency modeling capabilities.

**Graph Size** The original DAT requires a pre-defined graph size significantly larger than the source sequence length to model translation references, typically set $L = 8$. As illustrated in Fig 4, AEQA-DAT achieves peak performance with only $L = 3$, whereas DAT and DAT* exhibit gradual performance improvements as L increases, yet their final performance remains inferior to that of AEQA-DAT at $L = 3$. Overall, AEQA enables DAT to streamline the graph size, thereby enhancing training efficiency.

**Different Lengths** To analyze the impact of varying lengths on model performance, we categorized references into different length intervals and evaluated translation quality within each interval. As shown in Fig 5, AEQA-NAT demonstrates strong performance across all reference sequence length

*Table 5.* Results of different translation models on WMT16 EN→RO. We encompass a wide range of metrics including rule-based metrics (BLEU and chrf) and model-based metrics (COMET and BLEURT).

| Methods | BLEU↑ | chrf↑ | COMET↑ | BLEURT↑ | Speedup↑ |
|---|---|---|---|---|---|
| **Raw Data** | | | | | |
| Vanilla NAT | 19.82 | 50.65 | 65.22 | 53.17 | 15.6x |
| GLAT | 26.38 | 56.34 | 73.53 | 62.89 | 14.2x |
| DAT | 32.71 | 57.28 | 76.08 | 66.45 | 13.9x |
| AEQA-NAT | 32.75 | 57.40 | 77.12 | 67.64 | 17.0x |
| **Distillation** | | | | | |
| Vanilla NAT | 27.29 | 56.38 | 72.16 | 62.01 | 15.6x |
| GLAT | 31.19 | 57.07 | 75.23 | 65.17 | 14.2x |
| DAT | 32.79 | 57.81 | 76.52 | 66.84 | 13.9x |
| AEQA-NAT | 32.80 | 57.80 | 77.01 | 66.92 | 17.0x |

intervals, achieving results comparable to state-of-the-art NAT models. For AEQA-DAT, its translation quality surpasses that of DAT in every interval, further validating the effectiveness and stability of AEQA.

**N-gram Repetition** We evaluated the ability of AEQA to handle n-gram repetition. As illustrated in the Fig 6, AEQA-DAT demonstrates a significant advantage over the DAT models in reducing n-gram repetition. This improvement can be attributed to AEQA's reduction of input length, where the graph size decreases from 8 to 3, thereby streamlining the number of references on the directed acyclic graph. This effect is particularly pronounced for n-grams with sizes smaller than 4. Compared to all NATs, AEQA-NAT consistently maintains a lower n-gram repetition count, demonstrating its robust capability to handle multimodal challenges effectively.

**Sampling Ratio in Inference** The results across varying sampling rates demonstrate that AEQA-NAT maintains robust performance under diverse data sampling conditions during inference, as illustrated in Table 3. This underscores that the SQS effectively addresses the limitations of traditional methods reliant on explicit target word dependencies for token-level relationship modeling, thereby affirming the efficacy of NATs in leveraging semantic consistency spaces. For WMT14 En-De, model performance improves progressively as the sampling rate $\lambda$ increases from 0.1 to 0.4, reaching its peak at $\lambda = 0.4$. Beyond this point, performance declines, indicating that excessively high sampling rates may introduce redundancy or noise. A comparable pattern is observed for WMT17 Zh-En, with optimal performance achieved at $\lambda = 0.5$. These findings suggest that optimal sampling rates are dataset-dependent, emphasizing the necessity of tailoring $\lambda$ to specific data distributions.

**Multiple References** To assess the translation quality of AEQA-NAT from a multimodal perspective, we evaluated

its performance on the dataset[6] released by Ott et al. 2018, which includes ten reference translations for each of the 500 sentences from the WMT14 EN-DE test set. As demonstrated in Table 4, AEQA-NAT significantly surpasses other models in multi-reference translation tasks on raw data. This superiority arises from its ability to effectively capture multimodal references, allowing it to fully exploit its robust diversity generation capability in handling "one-to-many" mapping relationships. Notably, while all models show performance gains after distillation, AEQA-NAT's advantage becomes less pronounced compared to its performance on raw data. This further underscores that AEQA-NAT substantially reduces its dependence on knowledge distillation.

**Comprehensive Performance Evaluation** To comprehensively evaluate the performance of AEQA-NAT, we conducted assessments across multiple key benchmarks. As shown in Table 5, on raw data, AEQA-NAT outperforms other NAT models on both rule-based metrics (BLEU and chrF) and model-based metrics (COMET and BLEURT), indicating its ability to generate more coherent and higher-quality translations. On distilled data, AEQA-NAT achieves superior performance on model-based metrics, suggesting that the introduction of the Semantic Quantization Space (SQS) better captures the semantic relationships between the source and target languages. This is because model-based metrics evaluate translation quality by measuring semantic relevance between sentences using parametric knowledge.

### 4.2. Ablation Study

**AEQA significantly enhances model performance.** As presented in Table 6, the incorporation of the Semantic Quantization Space (SQS) results in a performance gain exceeding 9 BLEU points (Line 1 vs. Line 2). This substantial improvement underscores the critical role of AEQA training in enhancing the model's capacity to capture data distribution features. Furthermore, we observe a consistent trend of improved translation quality with an increase in the categorical number $K$ of the SQS, with optimal performance attained at $K = 2048$. Specifically, as demonstrated in Line 8 and Line 9, the BLEU scores achieve 25.14 and 25.96 under distinct sampling strategies, respectively. These results highlight the importance of selecting an appropriate $K$ value for maximizing model efficacy.

**Influence of Sampling Strategies.** The experimental results reveal that Adaptive Sampling exhibits a marginally better performance trend than Uniform Sampling under identical $K$ value configurations. At $K = 512$, the performance gap between the two exceeds 1 BLEU point (Line 2 vs. Line 3). As $K$ increases, this gap progressively narrows, with BLEU score differences of 0.84, 0.82, and 0.68 for $K = 1024$,

---

6 https://github.com/facebookresearch/analyzing-uncertainty-nmt

*Table 6.* Ablation on WMT14 EN→DE test set with different combinations of techniques. "AR" denotes Aligned Reordering.

| Line | K | | | | Sampling | | AR | DSLP | BLEU |
|---|---|---|---|---|---|---|---|---|---|
| | 512 | 1024 | 2048 | 4096 | Uniform | Adaptive | | | |
| 1 | | | | | | | | | 10.84 |
| 2 | ✓ | | | | ✓ | | | | 19.93 |
| 3 | ✓ | | | | | ✓ | | | 20.62 |
| 4 | | ✓ | | | ✓ | | | | 21.85 |
| 5 | | ✓ | | | | ✓ | | | 22.69 |
| 6 | | | | ✓ | ✓ | | | | 22.83 |
| 7 | | | | ✓ | | ✓ | | | 23.51 |
| 8 | | | ✓ | | ✓ | | | | 25.14 |
| 9 | | | ✓ | | | ✓ | | | 25.96 |
| 10 | | | ✓ | | | ✓ | ✓ | | 26.82 |
| 11 | | | ✓ | | | ✓ | ✓ | ✓ | **26.95** |

2048, and 4096, respectively. These findings indicate that the impact of sampling strategies on model performance decreases with larger $K$ values.

**Aligned Reordering** When evaluating only the predefined $K$ values and sampling methods without integrating AR and DSLP, the BLEU score is 25.96, as indicated in Line 9. With the inclusion of AR (Line 10), the BLEU score rises to 26.82. This improvement suggests that the AR mechanism further enhances the model's translation performance, potentially by refining syntactic structures or improving semantic alignment, thereby elevating the quality of the generated translations.

## 5. Related Work

The optimization strategies for NAT systems can be broadly categorized into two dimensions: **1) Input-side enhancement**, which incorporates explicit target information to improve the model's ability to capture data distributions, as exemplified by techniques as Conditional Masked Language Model (CMLM, Ghazvininejad et al. 2019; Du et al. 2021; Cheng & Zhang 2022; Shao et al. 2023) and Glancing Transformer (GLAT, Qian et al. 2021; Bao et al. 2022; Schmidt et al. 2022; An et al. 2023); and **2) Target-side optimization**, which modifies learning objectives to alleviate training difficulties, including methods like Aligned Cross Entropy (AXE, Ghazvininejad et al. 2020a), Order-Agnostic Cross Entropy (OAXE, Du et al. 2021), Multi-Granularity Optimization (MgMO, Li et al. 2022) and Non-Monotonic Latent Alignments (NMLA, Shao & Feng 2022). Target-side optimization techniques, such as AXE and OAXE, attempt to accommodate NAT training by relaxing strict word order alignment. Furthermore, several significant Aligned Reordering approaches have been applied to NAT. Reorder-NAT (Ran et al., 2021) incorporates a dedicated reordering module to explicitly model word rearrangement information during decoding. AligNART (Song et al., 2021)

achieves one-to-one mapping between encoder hidden representations and target information through alignment estimation, thereby reducing the modality of target distributions. AEQA-NAT leverages the synergistic effect of multiple loss components to jointly consider both the input and decoding phases, guiding the model to adaptively align the source and target texts.

## 6. Conclusion

In this work, we propose the AEQA-NAT training framework for non-autoregressive translation. By introducing a semantic quantization space, AEQA-NAT eliminates reliance on target information and effectively bridges the training-inference gap in NAT. Experimental results demonstrate that AEQA-NAT achieves state-of-the-art performance among fully non-autoregressive models across multiple translation benchmarks, while maintaining decoding speeds comparable to Vanilla NAT. Furthermore, our approach enhances the ability of NAT models to learn from raw data distributions, reducing the performance gap between raw data and knowledge distillation to 0.29 BLEU score.

## Impact Statement

This work presents a novel approach to bridge the training-inference gap in NAT models. By reducing the dependency on knowledge distillation, our method enhances the efficiency and scalability of NAT training. The proposed framework has the potential to enable more effective and efficient NAT deployment in real-world applications, contributing to the advancement of machine translation and other sequence generation tasks.

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

# A. Pre-aligned SQS

## A.1. VAE-based NAT

Latent Transformer (LT, Kaiser et al. 2018) is the first to use the vector quantization (VQ) technique to improve NAT learning. During inference, VAE-based NATs first predict the latent variable, then non-autoregressively produce the entire target sequence $y$ conditioned on the latent sequence (Kaiser et al., 2018; Shu et al., 2020; Bao et al., 2021; 2022). In contrast, we do not introduce an additional network to model the distributions of latent variables. We design NAT-based Semantic Quantization Space (SQS) for aligning discrete representations of source and target languages. Specifically, AEQA-NAT differs from latent-GLAT in several key aspects:

- AEQA-NAT introduces a novel Semantic Quantization Space (SQS) that jointly models discrete latent variables for both source and target texts during pre-alignment. This bilateral alignment mechanism, driven by the collaborative effect of multiple loss terms, establishes a unified semantic quantification space, ensuring cross-lingual consistency between training and inference. In contrast, latent-GLAT adopts a unilateral approach, encoding only target-side information and requiring an auxiliary network for latent variable prediction during inference.

- While latent-GLAT employs glancing training—optimizing masked token reconstruction by training on both latent variables and explicit tokens—it directly predicts complete translations during inference. Conversely, AEQA-NAT maintains training-inference parity by directly generating full translations in both phases, eliminating architectural discrepancies.

## A.2. Optimizing Semantic Quantization Alignment

To ensure effective alignment, we design a fine-tuning task on the pre-trained multilingual model mBART (Liu et al., 2020) for optimizing the pre-aligned SQS, as shown in Fig 2. For the sake of brevity, only the details related to the SQS are presented here. Given a language pair $(X, Y)$, and through the mBART encoder, we obtain the intermediate representation $\mathbf{H}_x$. Note that SQS contains the pre-aligned embeddings $z_q(\mathbf{H}_x)$ and $z_q(\mathbf{H}_y)$. To ensure translation quality, we prefer $z_q(\mathbf{H}_y)$ to keep the original embedding position and $z_q(\mathbf{H}_x)$ to cluster to $z_q(\mathbf{H}_y)$. We define the training objective for the alignment of the semantic quantization $\mathcal{L}_{\text{AL}}$ as

$$\mathcal{L}_{\text{AL}} = \mathcal{L}_{\text{a}} + \alpha \mathcal{L}_{\text{x}} \tag{16}$$

$$\mathcal{L}_{\text{a}} = \| z_{\bar{q}}(\mathbf{H}_x) - \text{sg}(z_{\bar{q}}(\mathbf{H}_y)) \|_2^2 \tag{17}$$

$$\mathcal{L}_{\text{x}} = \| \mathbf{H}_x - \text{sg}(z_q(\mathbf{H}_x)) \|_2^2 \tag{18}$$

where $\alpha$ is the hyperparameter to control the effect of $\mathcal{L}_x$, $\text{sg}(\cdot)$ is the stop-gradient operation and $\mathbf{H}_y$ is the intermediate representation output by the encoder for the target text, having the same dimension as $z_q(\mathbf{H}_y)$. Note that $z_{\bar{q}}(\mathbf{H}_x)$ is calculated as $\frac{1}{N_x} \sum_{i=1}^{N_x} z_q(h_{x,i})$ and $z_{\bar{q}}(\mathbf{H}_y)$ is calculated as $\frac{1}{N_y} \sum_{j=1}^{N_y} z_q(h_{y,j})$, which represent the closet embedding optima for $X$ and $Y$ in SQS, respectively.

The specific meaning of each loss term is as follows:

- $\mathcal{L}_{\text{a}}$: Ensures that the representation $z_{\bar{q}}(\mathbf{H}_x)$ of the source text is semantically aligned with the representation $z_{\bar{q}}(\mathbf{H}_y)$ of the target text quantization. This helps the model establish accurate semantic associations between different texts, enhancing the accuracy of cross-text semantic transfer and facilitating the subsequent generation of semantically coherent texts.

- $\mathcal{L}_{\text{x}}$: Maintains a reasonable relationship between the original intermediate representation $\mathbf{H}_x$ of the source text and its corresponding quantized representation $z_q(\mathbf{H}_x)$(after stop-gradient processing), preventing the quantization process from excessively distorting the original semantics and ensuring that the quantized representation $z_q(\mathbf{H}_x)$ can still effectively carry the key semantic information of the source text.

# B. Knowledge Distillation

Existing NAT techniques rely on knowledge distillation (Kim & Rush, 2016; Zhou et al., 2020) to mitigate the multimodality problem. Formally, the distillation data is a mixture of marginal distribution approximating the original data distribution and

the teacher model's distribution

$$\tilde{y} = \arg\max_{\hat{y}\in\mathcal{T}} sim(\hat{y}, y)q_\theta(\hat{y}|x)$$
$$\approx \arg\max_{y\in\mathcal{T}_K} sim(\hat{y}, y) \tag{19}$$

where $sim$ is a function measuring closeness by sentence-level BLEU (Chen & Cherry, 2014), $\mathcal{T}_K$ is the K-best list from beam search and $q_\theta$ is the teacher model. The distribution learned by the student model should match the mixture distribution

$$\mathcal{D}_{\text{KD}}(x, \tilde{y}) \sim (1-\alpha)\mathcal{D}_{\text{data}}(x, y) + \alpha q_\theta(\hat{y}|x) \tag{20}$$

Notably, the argmax of this mixture distribution is unlikely to correspond to either $y$ (the ground truth of the original data) or $\hat{y}$ (the beam search output). Thus, distilled data serves as a compromise between real data and data suitable for NAT training, rather than representing *real sentences* applicable to real-world scenarios. Furthermore, distilled data has inherent limitations, such as its dependence on guidance from a teacher model, compared to the richness and potential of the original corpus.

## C. Details of Model Components

### C.1. Correcting Word Order via Alignment Probability Matrix

The alignment probability distribution matrix $\mathbf{A}$ is used to reorder the words in the input sequence $z_q(\mathbf{H}_m)$ to match the syntactic order of the target sequence. The steps are as follows:

**Matrix Computation**

The alignment probability matrix $\mathbf{A}$ is defined as:

$$\mathbf{A} = \text{softmax}\left(\mathbf{Y}z_q(\mathbf{H}_m)^T\right)$$

where each element $A_{ij}$ denotes the probability of aligning input word $h_j$ with target position $y_i$.

**Algorithm for Reordering**

---
**Algorithm 1** Aligned Reordering Process

---
**Require:** Alignment probability matrix $\mathbf{A}$, Input sequence $z_q(\mathbf{H}_m) = \{h_1, h_2, h_3, h_4, h_5\}$
**Ensure:** Reordered sequence $\mathbb{AR}(z_q(\mathbf{H}_m))$
 1: Initialize an empty list *reordered_sequence*.
 2: **for** each target position $i$ in the range 1 to $n$ (loop over target sequence positions $y_i$) **do**
 3:     Extract row $i$ from $\mathbf{A}$ to get alignment probabilities for $h_1, h_2, ..., h_m$.
 4:     Identify column index $j$ with the maximum probability in row $i$, which corresponds to the alignment of $y_i$ with $h_j$.
 5:     Place $h_j$ in position $i$ of *reordered_sequence*.
 6: **end for**
 7: **return** *reordered_sequence*

---

**Outcome**

The final *reordered_sequence* aligns with the syntactic structure of the ground truth sequence and is returned as the corrected order.

### C.2. Length Predictor

Length prediction can be formulated as a classification problem based on the intermediate representations generated by the encoder. Following (Lee et al., 2018b), we predict the target sequence length $m$. Given $m$, the decoder inputs $\mathbf{H}_m = h_{1:m}$

are computed using *Softcopy* (Li et al., 2018; Wei et al., 2019) as:

$$w_{i,j} = \text{softmax}(-|j - i|/\tau),$$
$$h_j = \sum_{i=0}^{T} w_{ij}h_{x,i}, \tag{21}$$

where the weight $w_{i,j}$ is determined by the positional distance between the source position $i$ and the target position $j$, and $\tau$ is a hyperparameter controlling the sharpness of the softmax function.

## D. DA-Transformer

The strict position alignment between predicted and target tokens in vanilla NAT models struggles to capture multimodal data distributions, often leading to generated tokens with mixed modality and repeated words. To address this, the directed acyclic decoder state length is upsampled to $L$, and $H_L = [h_1, h_2, \ldots, h_L]$ represents the decoder output hidden states, defined as vertex states. The probability of a path a is redefined as the position transition probability:

$$p_\theta(\text{a}|x) = \prod_i p_\theta(a_{i+1}|a_i, x) = \prod_i \mathbf{E}_{a_i, a_{i+1}}, \tag{22}$$

where $\mathbf{E} \in \mathbb{R}^{L \times L}$ is the transition matrix normalized by rows. Here, $\text{a} = \{a_1, a_2, \ldots, a_T\}$ is a set of decoder position indices sorted in ascending order, with size $|\text{a}| = n$, and $\Gamma$ contains all possible a with a size of $\binom{L}{n}$. For example, if the target length $n = 3$ and $L = 5$, $\Gamma$ contains $\binom{5}{3} = 10$ possible paths, such as $\text{a} \in \{0, 1, 2\}, \{0, 1, 3\}, \{2, 3, 4\}$. Specifically, the transition matrix is computed as:

$$\mathbf{E} = \text{softmax}\left(\frac{\mathbf{Q}\mathbf{K}^T}{\sqrt{d}}\right),$$
$$\mathbf{Q} = \mathbf{H}\mathbf{W}_\text{Q}, \quad \mathbf{K} = \mathbf{H}\mathbf{W}_\text{K}, \tag{23}$$

where $d$ is the hidden size, and $\mathbf{W}_\text{Q}$ and $\mathbf{W}_\text{K}$ are learnable parameters. DAT applies lower triangular masking to $\mathbf{E}$, restricting transitions to vertices with smaller indices to larger indices. Conditioned on the vertex states in $\mathbf{H}$ and the selected path a, the posterior probability of $y$ is calculated as:

$$p_\theta(y|\text{a}, x) = \prod_{i=1}^{T} p_\theta(y_i|a_i, x)$$
$$= \prod_{i=1}^{T} \text{softmax}(\mathbf{W}_\text{p}\mathbf{h}_{\text{a}_i}), \tag{24}$$

where $\mathbf{h}_{\text{a}_i}$ is the representation of the $i$-th vertex on the path a.

## E. More Analyses

### E.1. Sampling Rate in Training

*Table 7.* Results on IWSLT16 with decreasing sampling ratio.

| Sampling Ratio | $\lambda_s$ | $\lambda_e$ | BLEU |
|---|---|---|---|
| | 0.6 | 0 | 26.28 |
| | 0.6 | 0.1 | 26.75 |
| Decreasing | 0.6 | 0.2 | 29.83 |
| | 0.6 | 0.3 | 31.10 |
| | 0.6 | 0.4 | **32.34** |
| | 0.6 | 0.5 | 30.36 |

Drawing inspiration from the GLAT methodology, we reduce the sampling rate during training. In contrast to GLAT, our initial state begins at $\lambda_s = 0.6$, as AEQA does not depend on explicit target token inputs for dependency learning and

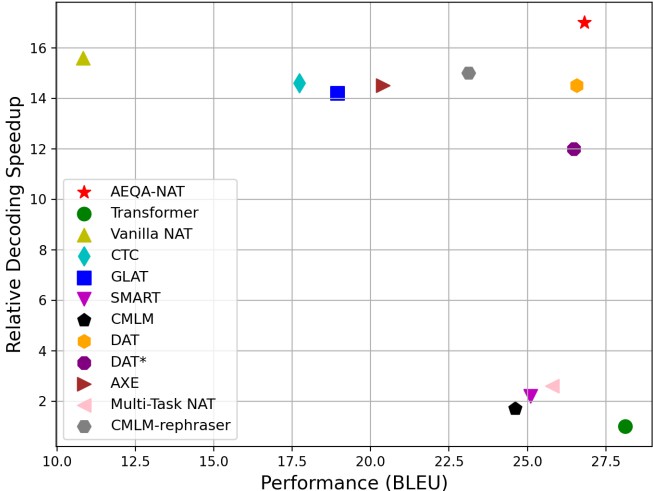

*Figure 7.* The tradeoff between Speedup and BLEU on WMT14 EN-DE.

necessitates richer semantic consistency information. Table 7 demonstrates the influence of different termination values $\lambda_e$ on model performance. As $\lambda_e$ increases from 0 to 0.4, the BLEU score exhibits a notable upward trend, reaching its peak of 32.34 at $\lambda = 0.4$. Therefore, we establish the training termination value at $\lambda_e = 0.4$ to fully exploit the model's capabilities.

### E.2. Tradeoff

As illustrated in Fig 7, AEQA-NAT demonstrates exceptional performance in terms of relative decoding speed, achieving a speedup that surpasses all other NATs. This result reinforces the advantage of NAT models in decoding speed, indicating that AEQA-NAT possesses significant superiority in decoding efficiency. Regarding translation quality, AEQA-NAT also excels, with its BLEU score surpassing that of other NAT models. This indicates that the translations generated by AEQA-NAT are closer in quality to the reference translations, further narrowing the performance gap with AT models. Overall, AEQA-NAT achieves an optimal performance-speed tradeoff.

