# OpenReview forum: "AEQA-NAT : Adaptive End-to-end Quantization Alignment Training Framework for Non-autoregressive Machine Translation"
_ICML.cc/2025/Conference — ICML 2025 poster_

### Official Review · Reviewer_rBqh · 2025-02-15

**Overall Recommendation:** 4

**Summary:**

The paper presents AEQA-NAT, a novel Non-Autoregressive Machine Translation (NAT) framework that introduces a Semantic Quantization Space (SQS) inspired by VQ-VAE. The key components include:
- Pre-aligned Semantic Quantization Space (SQS) leveraging mBART.
- Semantic Quantization Alignment Loss (LSQA) to enforce consistency.
- Aligned Reordering (AR) to improve syntactic alignment.

These innovations help AEQA-NAT mitigate the training-inference gap, enhancing translation quality while maintaining high decoding efficiency.

**Claims And Evidence:**

The paper claims that AEQA-NAT eliminates the training inference gap and achieves state-of-the-art performance among NAT models. These claims are supported by:
- Comprehensive experiments comparing AEQA-NAT against strong NAT baselines.
- Demonstrated performance improvements in BLEU scores, particularly on raw data.
- Reduced dependency on knowledge distillation, indicating improved generalization.

However, AEQA-NAT still lags behind the basic Autoregressive baseline, Transformer, in translation quality.

**Essential References Not Discussed:**

N/A

**Experimental Designs Or Analyses:**

The experiments are extensive across different WMT benchmarks and include ablation studies on SQS size, sampling strategies, and length effects.

**Methods And Evaluation Criteria:**

The methodology is well-motivated, combining quantization-based semantic alignment with adaptive reordering to address NAT’s dependency modeling issues. The evaluation on standard benchmarks (WMT14, WMT16, etc.) is appropriate, and comparisons with strong NAT baselines validate its performance.

One concern I have with the method is its reliance on Aligned Reordering (AR), especially for low-resource languages where alignment might be less reliable.

**Other Comments Or Suggestions:**

N/A

**Other Strengths And Weaknesses:**

N/A

**Questions For Authors:**

To use the aligned reordering technique, what is the computational cost? Do you compute the alignment with some aligner model on the fly during inference?

Previous work address the multimodality problem in NAT by applying knowledge distillation or using DAG (as in DAT) to generate coherent translation. In your work, I guess your semantic quantization and aligned reordering do a similar thing right? Can you provide more intuition behind this?

**Relation To Broader Scientific Literature:**

The paper builds upon previous NAT improvements, particularly knowledge distillation-based approaches and Directed Acyclic Transformers (DAT). Instead of distillation, AEQA-NAT introduces a semantic quantization approach similar to latent variable modeling in VQ-VAE. The work is well-positioned within the broader trend of improving NAT dependency modeling, offering an alternative path to addressing multimodality beyond KD and DAG-based methods.

**Theoretical Claims:**

N/A (this is an empirical paper)

---

> ### Author Rebuttal · Authors · 2025-03-31
>
> Thank you for your comments. Your feedback has been very helpful to us.
>
> 1. The cost of aligned reordering is $O(n \cdot m \cdot d)$, and we do not use an additional aligner model during the inference phase. We appreciate your interest in the intuition behind our method. When we recognized the existence of the training-inference gap in MLM-based NAT systems, we naturally hypothesized that this gap would hinder the performance of NAT, and our experimental results confirmed this hypothesis. Inspired by VQ-VAE, we quantize the semantic representation vectors, which forces the model to learn more compact semantic consistency representations. This not only facilitates bridging the training-inference gap but also mitigates the multimodal distribution characteristics in language mapping. Based on the discretized semantic representations, we model the word order correspondence through the aligned reordering mechanism, ensuring the syntactical accuracy of the discrete representations, which further alleviates the multimodal problem in NAT.
>
> 2. This work conducts a detailed study on typical datasets widely used in NAT systems (such as WMT ’14, ’16, etc.). We appreciate your mention of low-resource languages in relation to the reliability of our method, and this will be one of the key areas of focus in our future work.

---

### Official Review · Reviewer_EKGr · 2025-03-14

**Overall Recommendation:** 4

**Summary:**

Non-autoregressive transformers (NATs) are attractive due to computational efficiency for machine translation workloads. However, existing approaches fail to completely close training-inference mismatches for these systems. This work proposes Adaptive End-to-End Quantization Alignment Training for NATs (AEQA-NATs) to reduce this gap, optimizing for a few novel training goals and better leveraging a joint semantic embedding space. When successfully applied, AEQA-NATs achieve the fastest decoding throughput of any NAT model and see improvements in minimizing the BLEU score gap between autoregressive and non-autoregressive transformers.

**Claims And Evidence:**

The core claims of this work are clearly outlined in earlier sections and then supported by the methodology and data in Section 3. The ablation results later in this work serve to further support the core claims of this paper.

**Essential References Not Discussed:**

n/a

**Experimental Designs Or Analyses:**

The experimental design appears fairly standard for machine translation works. As mentioned before, typical metrics are employed. A range of similar NAT models are compared against in a seemingly comprehensive manner. Additionally, a thorough and credible ablation study is conducted.

**Methods And Evaluation Criteria:**

Chosen evaluation criteria seem fairly standard for this application space. BLEU, COMET, and similar, typical translation metrics are employed on typical datasets (WMT ’14, ’17, etc.).

**Other Comments Or Suggestions:**

The series of expressions under Section 2.3 is not well set-up and is an area for improvement on revision passes. While clear upon reinspection, it is dense and a little difficult to initially parse.

Typos: entend -> extend in Section 3 under Implementation

**Other Strengths And Weaknesses:**

Regarding strengths, the paper is very will written, which is particularly helpful as it is fairly information-dense. The proposed approach makes intuitive sense and the shared SQS is a sensible improvement. Additionally, the obtained results make a strong case for the viability of the proposed approach. When it comes to weaknesses, it is a bit unclear how stable the provided results are (i.e. do different training runs result in significant differences in end evaluations). Additionally, using the original transformer architecture as an autoregressive baseline is dubious, although this obviously does not impact the significance of demonstrated improvement upon NAT methods.

**Questions For Authors:**

Why was the original transformer architecture chosen as an autoregressive baselines? With nearly a decade of architectural improvements, surely a more modern baseline could have been chosen for a more reasonable comparison.

**Relation To Broader Scientific Literature:**

Based on the results observed in this work, the described approach is largely a step forward for machine translation NATs. While somewhat inspired by certain VAEs, the significance of this work appears to be constrained only to efficient machine translation.

**Theoretical Claims:**

This is largely an empirical work, no significant theoretical claims are made that need to be substantiated.

---

> ### Author Rebuttal · Authors · 2025-03-31
>
> Thank you for your valuable comments. Your feedback is important to us.
> 1. In response to your suggestion, we have added further descriptions to the sequence of expressions in Section 2.3 to more clearly explain the data flow process:
> - The source text $(x_1,x_2, \dots, x_n)$ is input into the encoder $f_{\text{enc}}$ to obtain the hidden representations $(h_{x,1},h_{x,2}, \dots, h_{x,n})$.
> - The length prediction module and the Softcopy mechanism transform the hidden representations into intermediate representations $(h_1,h_2, \dots, h_m)$.
> - The semantic consistency representations $(z_q(h_1),z_q(h_2), \dots, z_q(h_m))$ are obtained by querying the Semantic Quantization Space (SQS).
> - Finally, the semantic consistency representations $(z_q(h_1),z_q(h_2), \dots, z_q(h_m))$ and the encoder's hidden representations
> $(h_{x,1},h_{x,2}, \dots, h_{x,n})$ are fed into the decoder $f_{\text{dec}}$ for the decoding process.
> 2. Regarding your suggestion to "choose a more modern and improved autoregressive Transformer architecture as the autoregressive (AT) baseline," although existing NAT systems have made significant progress in translation quality, their overall performance still lags behind that of the vanilla Transformer. Therefore, the vast majority of NAT systems use the original Transformer architecture as the baseline for AT systems for comparison. We have adopted this approach as well. Your suggestion is valuable and will be an important direction for future exploration.

---

### Official Review · Reviewer_ifHW · 2025-03-17

**Overall Recommendation:** 2

**Summary:**

This paper works on non-autoregressive machine translation. It bridges the gap by introducing the latent variables and applying glancing training over the latent codes. In addition, order alignment of latent code is also introduced. The empirical results are very good.

**Claims And Evidence:**

The main claim is that the proposed method bridges the training-inference gap caused by Glat. The improved scores on standard datasets support this claim.

The main claim is not well-justified because the solution is a latent variable model, which naturally enhances the sequence modeling capability. It is not certain if glancing is useful at all in the proposed method.

My best understanding is that the proposed method is a well-designed latent variable model for NAR machine translation, but I think the motivation and the proposed method are not aligned.

Additional note: Glat, a training technique for many NAR systems, uses the training-inference gap. In fact, the gap is essential to maintaining performance (reducing it to 0 degrades performance). Thus, the gap may not be a drawback but rather a feature of Glat.

In fact, there is also a gap between the training and the inference in the proposed method. First, the glancing of the hidden codes are based on the groundtruth tokens, which are not available during the inference. The order alignment also needs ground truth during training.

**Essential References Not Discussed:**

The latent-Glat is closely related to this work. In fact, many aspects are the same:
1. Codebook, 2) glancing over latent codes, 3) non-autoregressive latent variables

Yu Bao, Hao Zhou, Shujian Huang, Dongqi Wang, Lihua Qian, Xinyu Dai, Jiajun Chen, Lei Li, latent-GLAT: Glancing at Latent Variables for Parallel Text Generation , ACL 2022

**Experimental Designs Or Analyses:**

As mentioned before, the proposed system introduces latent variables and claims to bridge the gap between the training and inference of glat training. This brings the concern that the main factor of the performance gain is from the latent variables, and has nothing to do with glat.
There should be at least a baseline line indicating that the glancing of latent code is not used (i.e., No SQ(x) in Eqns 11 and 12).

**Methods And Evaluation Criteria:**

This paper uses tokenized bleu for most datasets and includes chrf, COMET, and BLEURT for evaluating translation quality. It is generally valid and comprehensive.

However, using tokenized BLEU scores in NAR systems has been criticized over the years, and the paper should report SacreBLEU when possible (Tables 4 and 5).

**Other Comments Or Suggestions:**

The proposed system achieves good performance with relatively low FLOPs compared with models based on DAT or CTC. The authors may consider adding analysis to this regard.
The acronym DSLP is not explained nor cited.

**Other Strengths And Weaknesses:**

The hyper-parameters of glancing training are not explored, which is an issue as improving glancing is a major claim.

**Questions For Authors:**

When you compute the latency, what are the specs of the machine, including the GPU and CPU?

**Relation To Broader Scientific Literature:**

This paper relates to non-autoregressive machine translations, which involve methods like Glat, DAT, NPD, and DSLP.
In addition, VQ-VAE, which has broader applications, is also related.

**Theoretical Claims:**

There are no poofs in this work.

---

> ### Author Rebuttal · Authors · 2025-03-31
>
> Thank you for your comments.
> ## 1. Clarification on Motivation-Method Alignment
> To address your concern regarding the alignment between our motivation and the proposed method, we would like to clarify the following: We highly value the role of GLAT in NAR systems, which is why we chose it as our baseline model. Our intention to *bridge the training-inference gap* is not to reduce Glancing Sampling to zero (and in our work, we do not aim to eliminate Glancing Sampling), but rather to maintain consistent use of it during both training and inference. Specifically, we predict complete translations in both phases, as opposed to the mismatch seen in training  ($X+Y_{obs} \rightarrow Y_{mask}$) and inference  ($X \rightarrow Y$). Therefore, AEQA-NAT is designed to ensure consistency between training and inference, as detailed in Eq. (11) and Eq. (15). To summarize, our motivation is that the training-inference gap hinders NAT from fully realizing its potential, and we have empirically demonstrated this in Fig. 1. We propose AEQA-NAT, which introduces a semantic consistency space to semantically quantize and align the source and target texts. This approach preserves the MLM-based NAT modeling capability while overcoming the explicit reliance on target words, thereby achieving unified prediction of complete translations in both training and inference stages. Thus, our motivation is aligned with the proposed method.
>
> ## 2. Differentiation from Latent-GLAT
> We appreciate your attention to our methodological innovation and would like to clarify the significant differences between AEQA-NAT and Latent-GLAT (as cited on p.9). The following outlines the three key technical distinctions:
>
> - **Latent variable design**: AEQA-NAT introduces a novel *Semantic Quantization Space* (SQS) that jointly models discrete latent variables for both source and target texts during the pre-alignment phase. This bilateral alignment mechanism, driven by the collaborative effect of different loss terms, establishes a unified semantic quantification space that maintains cross-lingual consistency between training and inference phases. In contrast, latent-GLAT employs a unilateral approach that only encodes target-side information, requiring an additional network for latent variable prediction during inference.
> - **Training objective**: Latent-GLAT employs glancing training on both latent variables and explicit tokens during the training phase, optimizing for masked token reconstruction. During inference, it directly predicts complete translations. In contrast, AEQA-NAT directly predicts complete translations in both the training and inference phases, ensuring parity between training and inference.
> - **Architectural foundation**: As detailed in Appendix A.1 (p.13), AEQA-NAT diverges from VAE-based approaches that learn latent features through auxiliary networks. Instead of introducing additional networks to learn latent variables, our Semantic Quantization Space (SQS) **aligns vectorized discrete representations between source and target languages during the pre-alignment phase.**
>
> ## 3. Performance Attribution Analysis
> In response to your concern that "the main factor of the performance gain is from the latent variables, and has nothing to do with GLAT," our manuscript argues that the training paradigm of MLM-based NAT creates a training-inference gap. Therefore, our baseline primarily compares to MLM-based NAT systems. Thus, we focus on comparing the translation performance under Uniform sampling and Adaptive sampling methods, as shown in Table 6. We also present in Table 3 the impact of the sampling rate of GLAT during the inference phase on performance (with a fixed sampling rate during inference). The results demonstrate the effectiveness of the GLAT sampling method.
>
> ## 4. Hyperparameters of Glancing Training and Metric
> Regarding your comment on "the hyperparameters of glancing training not being explored," we have discussed the hyperparameters of glancing training in our settings:
> - Training-phase sampling rates: See Appendix E.1
> - Inference-phase sampling strategies: See Table 3
>
> For fair comparisons with previous work, we follow established practices from prior research and use sacreBLEU for WMT 17 EN-ZH, and tokenized BLEU for other benchmarks. The experimental results reported using sacreBLEU can be found in Table 3.
>
> ## 5. Other Comments and Questions
> Latency comparisons with vanilla Transformer were conducted on identical NVIDIA A40 GPU configurations. DSLP denotes deep supervision and additional layer-wise predictions. We appreciate your comment about the explanation and citation of DSLP, and we have now revised the manuscript to address this issue.

---

> > ### Comment · Reviewer_ifHW · 2025-04-07
> >
> > Thanks for the responses.
> >
> > For the revision: please include more discussion on Latent GLAT in the revision and VQ-VAE.
> >
> > Although  AEQA-NAT does not need to predict the codebook, which is shared for all samples and is still used during inference, AEQA-NAT is still a latent-variable model.  Therefore, I am not convinced that "AEQA-NAT diverges from VAE-based approaches.
> >
> > ### Regarding "Performance Attribution",
> > I do not disagree with the contribution of GLAT; rather, I am wondering which is the main contributor, using GLAT or being a latent model.
> >
> > 1. According to Table 6,
> >  - Having the latent variable (with K=4096) itself improves the performance from 10.84 to 25.14
> >  - Having GLAT only improves from 25.14 to 25.96.  (I am not sure if Table 3 uses AR, otherwise, 25.14 could be 26.82)
> > 2. In addition, Table 3 suggests the best performance without GLAT is 26.82 on WMT'14 EN-DE, which is the same as with GLAT (Line 10, DLSP doesn't count unless Table 3 uses DSLP).
> >
> > It is clear that being a latent mode contributes the most, and adding GLAT only makes a small difference (if any).
> >
> > ##  Regarding the "gap"
> > In addition, I am not sure if we can classify the proposed approach as addressing the training-inference gap because "MLM-based NAT creates a training-inference gap." Here is what I thought:
> > 1. CMLM (1 Iteration) < Vanilla NAT < GLAT: This tells us that the gap is not necessarily a bad thing for NAT.
> > 2. Without a gap in the proposed framework, the performance is actually worse (according to Table 3, where it is a clear trend that lambda = 0 is even worse).
> >
> > Overall, I do not doubt that the authors have proposed an efficient and strong NAT, but I don't think the motivation of this paper aligns with its solution.

---

> > > ### Author Response · Authors · 2025-04-08
> > >
> > > Thank you for your responses.
> > >
> > > We appreciate your comment on the efficiency and modeling capability demonstrated by the method we propose in this work.
> > > ## Response to Performance Attribution
> > > 1. We would like to clarify your comment that “Table 3 suggests the best performance without GLAT is…”. Please note that **Table 3 actually uses GLAT**, rather than being “**without GLAT**.” We would like to emphasize that this paper includes two GLAT processes:GLAT-Training and GLAT-Inference. In Table 3, GLAT is used in both the training and inference phases. In our setting, Glancing Training is the default, as can be understood from the context, specifically in Section 2.4, and Equations 11 and 12. To provide further clarification, we have expanded on the performance changes under different sampling ratios during inference, as shown in Table A:
> > > | Sampling ratio λ  | 0  | 0.1   | 0.2   | 0.3   | 0.4   | 0.5   | 0.6   | 0.7   | 0.8   | 0.9   | 1   |
> > > |------------------|-------|-------|-------|-------|-------|-------|-------|------|-------|-------|-------|
> > > | WMT14 EN-DE     | 18.10  | 18.34 | 20.71 | 24.53 | **26.82** | 25.64 | 23.87 | 23.08   | 22.42   | 22.57   | 21.63  |
> > >
> > >
> > > The results in **Table A reveal the contribution of GLAT during inference**. When $\lambda$ = 0, which is similar to existing GLAT-based methods, GLAT is used in the training phase, while the inference phase maintains the “training-inference gap.” The model’s performance on WMT14 EN-DE is 18.10, which is lower than the performance when λ takes other values [18.34, 26.82]. Therefore, the conclusion here should be: “Table 3 suggests that the best performance **with GLAT maintained during inference is 26.82 on WMT14 EN-DE**.”
> > >
> > > 2. Response to "Which is the Main Contributor (GLAT or Latent Model)"
> > >
> > > We have expanded the ablation study (as shown in Table B) to address this question:
> > > | Line | K = 2048 | GLAT-Train | GLAT-Inference | BLEU  |
> > > |------|----------|---------------|----------------|-------|
> > > | 1    |              |               |               | 10.84 |
> > > | 2    | ✓          |               |               | 14.57 |
> > > | 3    |   ✓        |        ✓    |               | 18.02 |
> > > | 4    |    ✓       | ✓           |      ✓      | 25.96 |
> > >
> > > The results in Table B provide an intuitive demonstration of the contribution of GLAT (at different stages). Please note that in our setting, GLAT samples the latent variable, so it cannot be applied without including the latent variable (see Eq. 12 and its explanation on Page 4, “Note that what we…”). According to Table B:
> > >
> > > - The latent variable itself (with K=2048) improves performance from 10.84 to 14.57(not 25.14).
> > > - With the addition of SQS, applying GLAT during training further improves the performance from 14.57 to 18.02.
> > > - Most importantly, after applying GLAT during inference, the performance increases to 25.96, as shown in Line 4. The core claim of our paper is that, when comparing Line 3 (without GLAT in inference) and Line 4 (with GLAT in inference) in Table B, a significant performance improvement is observed (from 18.02 to 25.96).
> > >
> > > Based on the above, both the SQS and GLAT during the training and inference phases significantly improve the performance of AEQA-NAT, **with the most notable performance improvement occurring during inference with GLAT, achieving a 7.94 BLEU increase**.
> > >
> > > 3. We would like to clarify the motivation behind our use of the latent variable. Upon discovering the training-inference gap in MLM-based NAT systems, we naturally hypothesized that this gap would limit the full potential of NAT. Our experimental results confirm this hypothesis. Inspired by VQ-VAE, we adopted semantic vector quantization (latent variable) as a representation of semantic consistency, which satisfies the needs of both the training and inference processes for NAT. Thus, we introduced this method to bridge the training-inference gap.
> > >
> > > ## Response to Gap Discussion
> > > 1. Regarding your comment, “CMLM < Vanilla NAT < GLAT, this tells us that the gap is not necessarily a bad thing for NAT,” if I understand correctly, you are suggesting that both CMLM and GLAT exhibit this gap, but GLAT outperforms Vanilla NAT, hence “the gap is not necessarily a bad thing.” We would like to clarify once again, as we previously mentioned in the rebuttal ("Clarification on Motivation-Method Alignment") and in the paper (Fig. 1, Eq. 12, and Eq. 15):
> > >
> > > - Our goal is not to reduce glancing sampling to zero.
> > > - The training-inference gap we refer to in MLM-based NAT systems pertains to the mismatch between the training process ($X+Y_{obs} \rightarrow Y_{mask}$) and the inference process ($X \rightarrow Y$).
> > > - Our proposed training-inference consistency refers to maintaining consistency between the training phase ($X+SQ(X) \rightarrow Y$) and inference phase ($X+SQ(X) \rightarrow Y$), as empirically demonstrated in Line 4 of Table B.
> > >
> > > 2. We appreciate your comment and will include a discussion on latent-GLAT in the revised version of the paper.

---

### Official Review · Reviewer_e42P · 2025-03-18

**Overall Recommendation:** 4

**Summary:**

This paper argues that there is a training-inference gap in Non-autoregressive Transformers (NATs), where NATs sample target words during training to enhance input but have no access to target information during inference. To address this, they propose an Adaptive End-to-end Quantization Alignment (AEQA) training framework, which introduces a semantic consistency space to eliminate the need for target information during inference. Experimental results demonstrate the effectiveness of AEQA, especially on raw data.

**Claims And Evidence:**

No. The claim "Experimental results demonstrate that our method achieves state-of-the-art performance among fully NAT models on major WMT benchmarks" is problematic. The state-of-the-art method FA-DAT [1] is ignored, which outperforms AEQA-NAT on raw data.

[1] Fuzzy Alignments in Directed Acyclic Graph for Non-Autoregressive Machine Translation, ICLR 2023.

**Essential References Not Discussed:**

AEQA-NAT employs an aligned reordering mechanism to align the word order of source and target sentences. The concept of aligned reordering is not new in NAT [1,2], but they are not cited.

[1] AligNART: Non-autoregressive Neural Machine Translation by Jointly Learning to Estimate Alignment and Translate, EMNLP 2021.
[2] Guiding Non-Autoregressive Neural Machine Translation Decoding with Reordering Information, AAAI 2021.

The state-of-the-art fully-NAT method on raw data is ignored [3].

[3] Fuzzy Alignments in Directed Acyclic Graph for Non-Autoregressive Machine Translation, ICLR 2023.

**Experimental Designs Or Analyses:**

Yes.

**Methods And Evaluation Criteria:**

Yes. The evaluation covers a wide range of benchmark datasets and metrics.

**Other Comments Or Suggestions:**

The abbreviation "AR" for "aligned reordering" can be misleading, as AR usually refers to "autoregressive". I recommend using a different abbreviation or simply labeling it as "w/ reordering" in Table 1.

**Other Strengths And Weaknesses:**

Pros:
1. This article identifies the training-inference gap issue in NAT, which makes sense and has been overlooked in previous research. The proposed Quantization Alignment method cleverly addresses this issue.
2. AEQA-NAT achieves impressive experimental results, especially in enabling vanilla NAT w/o distillation to perform nearly as well as the autoregressive baseline.
3. Both the experiments and analyses are very comprehensive.

Cons:
I do not find significant weaknesses of this paper.

**Questions For Authors:**

None

**Relation To Broader Scientific Literature:**

Currently, there is limited interest on non-autoregressive machine translation, yet the studies conducted in this area have the potential to inspire broader fields of research, such as the LLM community, which primarily relies on the next token prediction mechanism.

**Theoretical Claims:**

There are no theoretical claims.

---

> ### Author Rebuttal · Authors · 2025-03-31
>
> Thank you for your comments. Your suggestions have been very helpful to us.
> 1. In the revised manuscript, we have added citations to the relevant literature you mentioned and revised the conclusions in both the abstract and the experimental sections accordingly.
> 2. In response to your suggestion, we have changed "AR" to "w/ reordering" in Table 1 to avoid any potential confusion.

---

### Decision · Program_Chairs · 2025-05-01

**Decision:**

Accept (poster)

**Comment:**

The paper presents AEQA-NAT, a novel Non-Autoregressive Machine Translation (NAT) framework that introduces a Semantic Quantization Space (SQS) inspired by VQ-VAE. The key components include:

Pre-aligned Semantic Quantization Space (SQS) leveraging mBART.
Semantic Quantization Alignment Loss (LSQA) to enforce consistency.
Aligned Reordering (AR) to improve syntactic alignment.
These innovations help AEQA-NAT mitigate the training-inference gap, enhancing translation quality while maintaining high decoding efficiency.


This article identifies the training-inference gap issue in NAT, which makes sense and has been overlooked in previous research. The proposed Quantization Alignment method cleverly addresses this issue. AEQA-NAT achieves impressive experimental results, especially in enabling vanilla NAT w/o distillation to perform nearly as well as the autoregressive baseline.

For the revision: please include more discussion on Latent GLAT in the revision and VQ-VAE.